# PROVABLY EFFICIENT MULTI-TASK REINFORCEMENT LEARNING IN LARGE STATE SPACES

## ABSTRACT

We study multi-task Reinforcement Learning where shared knowledge among different environments is distilled to enable scalable generalization to a variety of problem instances. In the context of general function approximation, Markov Decision Process (MDP) with low Bilinear rank encapsulates a wide range of structural conditions that permit polynomial sample complexity in large state spaces, where the Bellman errors are related to bilinear forms of features wi 'th low intrinsic dimensions. To achieve multi-task learning in MDPs, we propose online representation learning algorithms to capture the shared features in the different task-specific bilinear forms. We show that in the presence of low-rank structures in the features of the bilinear forms, the algorithms benefit from sample complexity improvements compared to single-task learning. Therefore, we achieve the first sample efficient multi-task reinforcement learning algorithm with general function approximation.

## 1 INTRODUCTION

The ability to capture informative representations that generalize among multiple tasks has become significant in various machine learning applications Li et al. (2014); Tsiakas et al. (2016); Baevski et al. (2019); D'Eramo et al. (2019); Kubota et al. (2020); Liu et al. (2019b). In the context of multi-task learning Caruana (1997); Baxter (2000); Yu et al. (2005), this ability is highly desirable and becomes vital to learn with fewer amount of samples than learning each single task individually. Representation learning Bengio et al. (2013) is a powerful approach for achieving such sample efficiency improvement.

This paper considers representation learning in Multitask Reinforcement Learning, an important class of meta Reinforcement Learning (meta-RL) Wang et al. (2016); Finn et al. (2017); Ritter et al. (2018). Reinforcement learning (RL) is a sequential decision-making problem where an agent aims to learn the optimal decisions by interacting with an unknown environment Sutton & Barto (2018). Empowered by representation learning with deep neural networks LeCun et al. (2015); Goodfellow et al. (2016), RL has achieved tremendous success in various real-world applications, such as Go Silver et al. (2016), Atari Mnih et al. (2013), Dota2 Berner et al. (2019), Texas Holdém poker Moravčík et al. (2017), and autonomous driving Shalev-Shwartz et al. (2016). Therefore, the benefit of using representation learning to extract joint feature embedding from different but related tasks emerged as an essential problem to investigate.

Specifically, this paper studies the problem of learning multiple RL problems jointly with the help of representation learning. Although multi-task learning in online-decision making problems has received increasing research interest Lazaric & Ghavamzadeh (2010); Mutti et al. (2021); Maurer et al. (2016); Qin et al. (2021); Yang et al. (2021); Hu et al. (2021), most existing works focus on tabular or linear models. Indeed, how general function approximations extrapolate across huge state spaces remains largely an open problem itself. Recently, Bilinear class Du et al. (2021) proposes a promising structural framework of generalization in reinforcement learning through the use of function approximation. Bilinear class postulates that the Bellman error can be related to a bilinear form depending on the hypothesis and captures nearly all existing function approximation models, e.g. Jin et al. (2020a); Zanette et al. (2020); Yang & Wang (2020); Jiang et al. (2017); Sun et al. (2019); Kakade et al. (2020); Agarwal et al. (2020). However, in the presence of shared information in the bilinear forms across multiple tasks, the Bilin-UCB proposed in Du et al. (2021) is not able to adapt

to such knowledge and challenges abound in adopting representation learning to find nearly-optimal policies with limited data.

In this paper, we give the first sample efficient algorithm of multi-task RL with *general function approximation* through the usage of representation learning in Bilinear class. In particular, to study representation learning, we propose Low-rank Multi-task Bilinear class—a structural framework that permits generalization both within and across tasks in multi-task RL. Specifically, such a model class specifies $M$ MDP instances, where $M > 0$ is a fixed integer and each one belongs to the Bilinear class Du et al. (2021), i.e., the Bellman error admits a low-rank factorization in $\mathbb{R}^d$. Since our multi-task setting have $M$ MDP instances, there are $M$ different features maps specified by the definition of Bilinear class, each corresponding to one MDP task and taking values in $\mathbb{R}^d$. We additionally assume that these $M$ features maps, when packed together as a matrix-valued mapping in $\mathbb{R}^{d \times M}$, has rank $k \ll d$. In other words, in Low-rank Multi-task Bilinear class, the bilinear form of each task possesses a low-dimensional task-specific feature and a shared representation.

Under this setting, it is desirable that the RL algorithm utilize the intrinsic low-dimensional structure to achieve an improved sample efficiency compared to solving each task separately. To this end, under the online setting where the agent learn from its past experiences without knowing the model, we design a sample efficient algorithm that provably finds nearly-optimal policies for all tasks. Our algorithm is based on the principle of Optimism in the Face of Uncertainty (OFU) which constructs an confidence region that contains the true hypothesis based on the historical data across the $M$ tasks, and then update the policy according to the most optimistic hypothesis within the confidence region. In particular, here the hypothesis can denote the true transition models or optimal value functions of these $M$ tasks. When constructing the confidence region, we explicitly utilize the low-dimensional structure by joint learning the task-specific features and shared representation via Empirical Risk Minimization (ERM) with multi-task data.

Moreover, as for planning, we find the hypothesis in the confidence region which leads to the highest aggregated value in these $M$ tasks. In the analysis, we show a concentration result where the estimation noise can be embedded into low dimension space and thus prove that our algorithm is able to find nearly-optimal policies within limited samples. Concretely, compared to learning each task separately using Bilin-UCB Du et al. (2021), an algorithm designed for Bilinear class without utilizing the shared representation, our algorithm enjoys a $(d/k)$-time improvement in the sample efficiency whenever the feature classes are small. To our best knowledge, our work seems to propose the first provably sample efficient multi-task RL algorithm with general function approximation.

**Notations**  Let $\mathbb{R}^d$ denote the $d$-dimensional space and $\mathbb{R}^{d \times k}$ denote the space of $d$-by-$k$ matrices in $\mathbb{R}$. The inner product of two vectors $x, y \in \mathbb{R}^d$ is denoted as $\langle x, y \rangle$. For sets $A_1, \ldots, A_n$, define $\otimes_{k \in [n]} A_k = A_1 \otimes \cdots \otimes A_k = \{(a_1, \ldots, a_n) : a_k \in A_k, k \in [n]\}$. Given scalars $a_1, \cdots, a_n$, let $a_{1:n}$ denote the vector $(a_1, \cdots, a_n)$. Also let $(a_\omega)_{\omega \in \Omega}$ denote the tuple consisting of $a_\omega$ where $\omega$ comes from a countable set $\Omega$. For variables $v_1, \ldots, v_k$, we denote by $v_{1:k}$ the $k$-tuple $(v_1, \ldots, v_k)$.

**Roadmap**  In Section 2 we introduce the basic problem setup and notations. In Section 3 we introduce Low-rank Multi-task Bilinear class—a framework that captures shared information in bilinear class. Next, we display the main algorithm of learning Low-rank Multi-task Bilinear class models, empowered by representation learning and optimism principle in decision making, in Section 4. We show the main theoretical result in Section 5 and the overview of techniques in Section 6. A couples of examples are given in Section A. We conclude with discussions of further directions.

## 1.1 RELATED WORK

**General function approximation in Reinforcement Learning**  Theoretical understanding of the sample complexity of RL with general function approximation seems relatively scarce. In recent years, there has been a surge of theoretical insights on linear function approximation and non-linear function approximations Jin et al. (2020b;c); Wang et al. (2021); Zanette et al. (2020); Agarwal et al. (2020); Kakade et al. (2020); Wen & Van Roy (2017); Dann et al. (2018); Du et al. (2019); Dong et al. (2020); Liu et al. (2019a); Wang et al. (2020a); Dong et al. (2021); Zhou et al. (2020); Yang et al. (2020); Jin et al. (2021a); Du et al. (2021). Among them, Bilinear class Du et al. (2021) is one of the most general framework.

**Multi-task learning and Meta learning** Meta Learning Thrun & Pratt (2012) and multi-task representation learning Caruana (1997) are important tools for capturing shared knowledge across tasks and achieving generalization to a new task. Theoretical analysis dated back to Baxter (2000); Maurer (2005); Yu et al. (2005). Maurer et al. (2016); Amit & Meir (2018); Konobeev et al. (2020) consider generalization errors averaged over the meta distribution under the assumption of shared distribution for sampling the source tasks. Recently, Du et al. (2020); Tripuraneni et al. (2020b;a); Chua et al. (2021) consider the diversity of task distributions and establish risk bounds based on learning shared representation across tasks and transferring to new tasks.

**Multi-task bandit and RL** In multi-task bandit and RL problems, theoretical analysis often postulates the existence of low-rank of sparsity structures in the representation Lazaric & Ghavamzadeh (2010); Brunskill & Li (2013); Calandriello et al. (2014); Mutti et al. (2021); Maurer et al. (2016); D'Eramo et al. (2019); Arora et al. (2020); Qin et al. (2021); Yang et al. (2021); Hu et al. (2021). The most related works are Hu et al. (2021); Yang et al. (2021) where the benefits of representation learning of linear bandits and linear MDPs are studied. However, it remains open whether multi-task RL can benefit from representation learning via general function approximation.

## 2 PRELIMINARIES

We consider learning a set of $M$ problem instances

$$\mathcal{P} = \{\mathcal{M}_m = (\mathcal{S}_m, \mathcal{A}_m, \{\mathbb{P}_{m,h}\}_{h=1}^H, r_m, s_{m,1})\}_{m \in [M]},$$

where each $\mathcal{M}_m \in \mathcal{P}$ denotes an episodic Markov Decision Process (MDP) in which $\mathcal{S}_m$ denotes the state space, $\mathcal{A}_m$ denotes the action set, $H$ denotes the number of time steps in each episode, $\{\mathbb{P}_{m,h}\}_{h=1}^H$ denotes the transition kernel, $r_m$ denotes the reward function, and $s_{m,1}$ denotes the fixed initial state. We assume $r_m \in (0, 1)$ without loss of generality.

For MDP $\mathcal{M}_m$, let $\mathbb{E}^m$ denote the expectation under $\{\mathbb{P}_{m,h}\}_{h=1}^H$. A (deterministic) policy $\pi_m$ is a length-$H$ sequence of functions $\pi_m = \{\pi_{m,h} : \mathcal{S}_m \mapsto \mathcal{A}_m\}_{h=1}^H$. To interact with $\mathcal{M}_m$, the agent starts at a fixed initial state $s_{m,1}$ and at each time step $h \in [H]$, it takes action $a_{m,h} \sim \pi_{m,h}$, receives reward $r_h(s_{m,h}, a_{m,h})$ and transits to $s_{m,h+1} \sim \mathbb{P}(\cdot|s_{m,h}, a_{m,h})$. Let $\mathbb{E}_{\pi_m}^m$ denote the expectation under MDP $\mathcal{M}_m$ and taking policy $\pi_m$. We use $o_{m,h} = (s_{m,h}, a_{m,h}, s_{m,h+1})$ to denote the history at $h$-th time step in MDP $\mathcal{M}_m$. Given a policy $\pi_m$, we define the value function $V_{m,h}^{\pi_m}(s)$ as the expected sum of reward under policy $\pi_m$ starting from $s_{m,h} = s$ at time step $h$:

$$V_{m,h}^{\pi_m}(s) := \mathbb{E}^m \left[ \sum_{t=h}^H r_m(s_{m,t}, a_{m,t}) | s_{m,h} = s \right].$$

Similarly, we define the Q-function $Q_{m,h}^{\pi_m}(s, a)$ as the the expected sum of reward taking action $a$ in state $s_{m,h} = s$ and then following $\pi_{m,h}$:

$$Q_{m,h}^{\pi_m}(s, a) = \mathbb{E}^m \left[ \sum_{t=h}^H r_m(s_{m,t}, a_{m,t}) | s_{m,h} = s, a_{m,h} = a \right].$$

The Bellman operator $\mathcal{T}_{m,h}$ applied to Q-function $Q : \mathcal{S}_m \times \mathcal{A}_m \mapsto \mathbb{R}$ is defined via

$$\mathcal{T}_{m,h}(Q)(s, a) := r_{m,h}(s, a) + \mathbb{E}_{s' \sim \mathbb{P}_{m,h}(\cdot|s,a)}[\max_{a'} Q(s', a')].$$

There exists an optimal policy $\pi_m^*$ that gives the optimal value function for all states, i.e. , $V_{m,h}^{\pi_m^*}(s) = \sup_\pi V_{m,h}^\pi(s)$ holds for all $h \in [H]$ and $s \in \mathcal{S}_m$. Therefore $Q_{m,h}^{\pi_m^*}$ satisfies the following Bellman optimality equations for all $s \in \mathcal{S}_m$, $a \in \mathcal{A}_m$ and $h \in [H]$:

$$Q_{m,h}^{\pi_m^*}(s, a) = \mathcal{T}_{m,h}(Q_{m,h+1}^{\pi_m^*})(s, a).$$

The agent aims at using fewer samples to find a set of policies $\{\pi_m\}_{m=1}^M$ such that

$$\sum_{m=1}^M V_{m,h}^{\pi_m^*}(s_1) - V_{m,h}^{\pi_m}(s_1) \leq \epsilon$$

holds with probability at least $1 - \delta$. In the following, we define the filtration $\mathcal{H}_t$ to be the $\sigma$-field induced by all the random variables up to round $t$.

## 3 MULTI-TASK BILINEAR CLASS

In general function approximation, a hypothesis class denoted by $\mathcal{G}$ is applied to address the set $\mathcal{P}$ of $M$ problem instances. Here, each hypothesis $g \in \mathcal{G}$ is a function approximation that captures the information of both task-specific information of all $M$ problem instances and the shared knowledge across tasks. For example, in model-based RL, $g$ may denote the transition models of the $M$ tasks, that is, $\{\mathbb{P}_{m,h}\}_{m=1,h=1}^{M,H}$. Whereas in model-free RL, $g$ can be the optimal value functions of the $M$ tasks, i.e., $\{V_{m,h}^{\pi_m^*}\}_{m=1,h=1}^{M,H}$. Using the notion of hypothesis $g \in \mathcal{G}$, we aim to cover a large class of MDP models studied under the function approximation setting. We will introduce concrete examples in Section A.

Under our setting, each hypothesis $g \in \mathcal{G}$ is associated with Q-functions $\{Q_{m,h,g}(\cdot,\cdot)\}_{m \in [M], h \in [H]}$ and value functions $\{V_{m,h,g}(\cdot)\}_{m \in [M], h \in [H]}$ in the $M$ tasks such that $V_{m,h,g}(\cdot) = \max_a Q_{m,h,g}(\cdot, a)$ holds for all $m \in [M]$ and $h \in [H]$.

Bilinear class Du et al. (2021) is a general framework that allows generalization in RL for a wide range of function approximators. In the following, we develop the low-rank multi-task bilinear framework so that it permits generalization in meta-RL across different tasks. The key intuition behind this framework is that it captures the common representation via low-rank structures in the features of task-specific bilinear forms.

**Definition 3.1** (Low-rank Multi-task Bilinear class). We say the following tuple

$$(\mathcal{G}, \{\mathcal{M}_m = (\mathcal{S}_m, \mathcal{A}_m, \mathbb{P}_m, r_m, s_{m,1})\}_{m \in [M]}, l_{m,h,f})$$

is a multi-task Bilinear class with rank $k$ if there exists $g^* \in \mathcal{G}$ such that $Q_{m,h,g^*} = Q_{m,h}^{\pi_m^*}$ and $V_{m,h,g^*} = V_{m,h}^{\pi_m^*}$ hold for all $m \in [M], h \in [H]$, and there exist functions $W_{m,h}^* : \mathcal{G} \mapsto \mathbb{R}^d$, $v_{m,h}^* : \mathcal{G} \mapsto \mathbb{R}^k$, $B_h^* : \mathcal{G} \mapsto \mathbb{R}^{d \times k}$, and $X_{m,h}^* : \mathcal{G} \mapsto \mathbb{R}^d$ ($d \gg k$) such that for each $g \in \mathcal{G}$, $m \in [M]$ and $h \in [H]$:

1. The features $W_{m,h}^*$ possess low rank structures:

$$W_{m,h}^*(g) = B_h^*(g)v_{m,h}^*(g). \tag{1}$$

2. We can upper bound the expected Bellman error as follows:

$$\left| \mathbb{E}_{a_{m,1:h} \sim \pi_{m,g,1:h}}^m [Q_{m,h,g}(s_{m,h}, a_{m,h}) - r_{m,h}(s_{m,h}, a_{m,h}) - V_{m,h,f}(s_{m,h+1})] \right|$$
$$\leq |\langle W_{m,h}^*(g) - W_{m,h}^*(g^*), X_{m,h}^*(g) \rangle|. \tag{2}$$

Here $\pi_{m,g,h}(s) = \arg\max_a Q_{m,h,g}(s, a)$ is the optimal policy in MDP $\mathcal{M}_m$ under hypothesis $g$.

3. For any $f \in \mathcal{G}$ there exist policy $\pi_{\text{est},m}(f)$ and discrepancy measure $l_{m,h,f}(o_{m,h}, g)$ that can be used for estimating $\langle v_{m,h}(g)B_h(g) - v_{m,h}(g^*)B_h(g^*), X_{m,h}(f) \rangle$ for any $g \in \mathcal{G}$, such that:

$$\mathbb{E}_{a_{m,1:h-1} \sim \pi_{m,f}, a_{m,h} \sim \pi_{\text{est},m}(f)}^m [l_{m,h,f}(o_{m,h}, g)] = \langle W_{m,h}^*(g) - W_{m,h}^*(g^*), X_{m,h}^*(f) \rangle, \tag{3}$$

where $o_{m,h} = (s_{m,h}, a_{m,h}, s_{m,h+1})$.

*Remark* 3.2. A few remarks are in order. First, when $d = k$ and $M = 1$, we recover the Bilinear class introduced in Du et al. (2021). Thus, our model can be viewed as an multi-task extension of Bilinear class with a shared low-dimensional representation. Second, here the hypothesis class $\mathcal{G}$ can either be model-based function approximation or value-based function approximation, as in either case the Q-functions $Q_{m,h,g}$ and the value functions $V_{m,h,g}$ are well defined. Third, the common representation $B_h^*(\cdot)$ captures the knowledge shared in all problem instances and enables multi-task learning with fewer samples. Finally, the discrepancy measure $l_{m,h,f}(o_{m,h}, \cdot)$ can be computed for all hypothesis $g$, an important property that facilitates data reusage as explained below. Examples of Low-rank Multi-task Bilinear class can be found in below and in Appendix A.

To understand this framework, we first notice that the expected Bellman errors of the following forms

$$\left| \mathbb{E}^m_{a_{m,1:h} \sim \pi_{m,g,1:h}} [Q_{m,h,g}(s_{m,h}, a_{m,h}) - r_{m,h}(s_{m,h}, a_{m,h}) - V_{m,h,f}(s_{m,h+1})] \right|$$

serve as upper bounds for the sub-optimality of policies $\pi_{m,g}$. Thus the Eq. (2) indicates that the sub-optimality of policies $\pi_{m,g}$ can be controlled if the feature $W^*_{m,h}(g)$ is close to the feature $W^*_{m,h}(g^*)$ corresponding to the best-in-class function approximator $g^*$. Eq. (3) further establishes the connection between the error $W^*_{m,h}(g) - W^*_{m,h}(g^*)$ and the discrepancy measure $l_{m,h,f}(o_{m,h}, g)$. One important observation is information sharing among the function class through the feature $X^*_{m,h}(f)$: given samples from a single function approximator $f$, the quantity

$$\mathbb{E}^m_{a_{m,1:h-1} \sim \pi_{m,f}, a_{m,h} \sim \pi_{\text{est},m}(f)} [l_{m,h,f}(o_{m,h}, g)]$$

can be estimated for all $g \in \mathcal{G}$. This thus allows for data reusage for each on-policy samples in our algorithm.

Eq. (1) postulates that the features $W^*_{m,h}(g)$ share a low-dimension structure. Note that this structure is not fixed for all function approximator $g$ — in fact, the common representation $B_h(g)$ may also be a function of $g$. This allows for much generality in multi-task RL models. We will show that the low-rank multi-task bilinear class cover many existing function approximation models in multi-task setting. As compared with a great number of multi-task/meta RL algorithms that use a single representation to solve multiple tasks, our function approximators allow for more generality and therefore handles more structural conditions of multi-task RL such as those studied in Yang et al. (2021) and Hu et al. (2021). Moreover, our algorithm reduces to using a single representation for the feature W in many special cases, for example when the feature $v^*_{m,h}$ is trivial. In this sense, our algorithm can be seen as encompassing the method of using a single feature for multiple tasks.

Now we give an example of low-rank multi-task bilinear framework. This example shows that the proposed framework captures the linear mixture model Modi et al. (2020) in multi-task RL where the mixing coefficients of different tasks lie in the same low-dimension space. First, we recall the definition of Linear Mixture Model in the following.

**Definition 3.3** (Linear Mixture Model). MDP $\mathcal{M} = (\mathcal{S}, \mathcal{A}, \{\mathbb{P}_h\}_{h=1}^H, r, s_1)$ is a linear mixture model if there exist known feature maps $\phi : \mathcal{S} \times \mathcal{A} \times \mathcal{S} \mapsto \mathbb{R}^d$ and $\psi : \mathcal{S} \times \mathcal{A} \mapsto \mathbb{R}^d$ and unknown $\theta^*_h \in \mathbb{R}^d, h \in [H]$ such that

$$\mathbb{P}_h(s, a, s') = \langle \theta^*_h, \phi(s, a, s') \rangle, \quad r_h(s, a) = \langle \theta^*_h, \psi(s, a) \rangle.$$

It is known that linear mixture model is Bilinear class with hypothesis $g = (\theta_1, \ldots, \theta_H)$ and

$$X_h(g) = \mathbb{E}_{\pi_g}[\psi(s_h, a_h) + \sum_{s' \in \mathcal{S}} \phi(s_h, a_h, s') V_{h+1,g}(s')], \quad W_h(g) = \theta_h.$$

The discrepancy measure can chosen as

$$l_f(o_h, g) = \langle \theta_h, \phi(s_h, a_h) + \sum_{s'} \psi(s_h, a_h, s') V_{h+1,f}(s') \rangle - (V_{h+1,f}(s_{h+1}) + r_h)$$

and the estimation policies can be chosen as $\pi_{\text{est}}(f) = \pi_f$. We consider learning the special case of Low-rank Multi-task Bilinear class where each MDP $\mathcal{M}_m$ is a linear mixture model with

$$\mathbb{P}_{m,h}(s, a, s') = \langle \theta^*_{m,h}, \phi_m(s, a, s') \rangle, \quad r_{m,h}(s, a) = \langle \theta^*_{m,h}, \psi_m(s, a) \rangle$$

and there exist $\nu^*_{m,h} \in \mathbb{R}^k, B^*_h \in \mathbb{R}^{d \times k}$ such that $\theta^*_{m,h} = B^*_h \nu^*_{m,h}$ for all $h \in [H], m \in [M]$. Then we let $g = (\nu_{m,h}, B_h)_{h \in [H], m \in [M]}$ and use the (fixed) features $v_{m,h}(g) = \nu_{m,h}, B_h(g) = B_h$. Notice that each $(\nu_{m,h}, B_h)_{h \in [H], m \in [M]}$ will define the expectation $\mathbb{E}^m_{\pi_{m,g}}$ (via $\mathbb{P}_{m,h}(s, a, s') = \langle \theta_{m,h}, \phi_m(s, a, s') \rangle$ with $\theta_{m,h} = B_h \nu_{m,h}$). Thus we can set the feature class $\mathcal{X}$ induced by $\mathcal{G}$ in which the feature $X_{m,h}(g)$ can be computed for each $g \in \mathcal{G}$ as follows:

$$X_{m,h}(g) = \mathbb{E}^m_{\pi_{m,g}}[\psi_m(s_{m,h}, a_{m,h}) + \sum_{s' \in \mathcal{S}_m} \phi_m(s_{m,h}, a_{m,h}, s') V_{m,h+1,g}(s')].$$

---

**Algorithm 1** Representation Learning in Low-rank Multi-task Bilinear class

---

**Input:** $\mathcal{V}, \mathcal{B}, \mathcal{X}$
**for** $t \leftarrow 1, \dots, T$ **do**
    Find $g^{(t)}$ as the solution of the optimization problem in Eq. (4)
    For $\forall h \in [H], m \in [M]$, sample $n_0$ times from $a_{m,1:h-1} \sim \pi_{m,g^{(t)}}, a_{m,h} \sim \pi_{m,\text{est}}$ and create
datasets $\mathcal{D}_{m,h}^{(t)}$.
**end for**
Let $t_0 \leftarrow \max_{t \in [T]} \sum_{m=1}^{M} V_{m,1}^{\pi_{m,g^{(t)}}}(s_1)$
**Return** $\pi^{g^{(t_0)}}$

---

## 4    REPRESENTATION LEARNING IN LOW-RANK MULTI-TASK BILINEAR CLASS

In this section we present the main algorithm to learn Low-rank Multi-task Bilinear class. First, we define in the following the feature classes to capture $v_{m,h}^*, B_h^*, X_{m,h}^*$ for $m \in [M], h \in [H]$.

**Definition 4.1** (Function approximations). We define feature class $\mathcal{V} = \mathcal{V}_1 \otimes \cdots \otimes \mathcal{V}_H$ where $\mathcal{V}_h \subset \{v : \mathcal{G} \mapsto \mathbb{R}^k\}^{\otimes M}, \forall h \in [H]$, representation class $\mathcal{B} = \mathcal{B}_1 \otimes \cdots \otimes \mathcal{B}_H$ where $\mathcal{B}_h \subset \{B : \mathcal{G} \mapsto \mathbb{R}^{d \times k}\}$, and feature class $\mathcal{X} = \mathcal{X}_1 \otimes \cdots \otimes \mathcal{X}_H$ where $\mathcal{X}_h \subset \{X : \mathcal{G} \mapsto \mathbb{R}^d\}^{\otimes M}, \forall h \in [H]$.

Here we assume that function classes $\mathcal{V}, \mathcal{B}$, and $\mathcal{X}$ completely captures the mappings specified in the multi-task Bilinear class. Therefore, given expressive $\mathcal{V}, \mathcal{B}, \mathcal{X}$ as inputs, we can learn the representations $v_{1:M,1:H} \in \mathcal{V}, B_{1:H} \in \mathcal{B}, X_{1:M,1:H} \in \mathcal{X}$ to approximate $v_{1:M,1:H}^*, B_{1:H}^*, X_{1:M,1:H}^*$ by minimizing some proper loss function.

Based on this assumption, we propose an algorithm based on the OFU principle while leveraging representation learning to improve sample efficiency. The procedure is shown in Algorithm 1. In particular, in the $t$-th iteration, an (optimistic) hypothesis $g^{(t)} \in \mathcal{G}$ is computed using Upper Confidence Bound (UCB) by finding a hypothesis that achieves the highest total value in these $M$ tasks. Specifically, consider the following constrained optimization problem:

$$g^{(t)} = \arg \max_{g \in \mathcal{G}^{(t)}} \sum_{m=1}^{M} V_{m,1,g}(s_{m,1}) \tag{4}$$

which maximizes the sum of candidate value functions $V_{m,1,g_i}(s_{m,1})$ subject to the constraint that the hypothesis $g^{(t)}$ belongs to the confidence set $\mathcal{G}^{(t)}$. As we will show in the proof, $\mathcal{G}^{(t)}$ contains the true hypothesis $g^*$ for all $t \in [T]$ with high probability. Thus, by solving (4), the sum of value functions of $g^{(t)}$ serves as an upper bound of that of $g^*$.

The key issue now is how the confidence set is chosen. In BiLin-UCB, the confidence set is chosen to contain all hypotheses that possess small values in the average of discrepancy measures across the available batch data. Since the discrepancy measures serve as unbiased estimates for the bilinear forms which upper bound the Bellman error, this confidence set essentially finds all hypotheses with low Bellman error. However, this approach fails to exploit the shared information among tasks. Instead, we make use of the feature classes $\mathcal{V}, \mathcal{B}, \mathcal{X}$ and learn the common representation $\{B_h\}_{h \in [H]}$ by Empirical Risk Minimization (ERM). For each hypothesis $g \in \mathcal{G}$ and $h \in [H]$, let $(v_{1:m,h}^{(g)}, B_h^{(g)}, X_{1:m,h}^{(g)}, \widetilde{g})$ be the solution of the following optimization problem:

$$(v_{1:m,h}^{(g)}, B_h^{(g)}, X_{1:m,h}^{(g)}, \widetilde{g}^{(g)}) = \underset{v_{1:m,h} \in \mathcal{V}_h, B_h \in \mathcal{B}_h, X_{1:m,h} \in \mathcal{X}_h, \widetilde{g} \in \mathcal{G}}{\arg \min}$$

$$\left\{ \sum_{\tau=1}^{t-1} \sum_{m=1}^{M} \left( \mathbb{E}_{(s,a,s') \sim \mathcal{D}_{m,h}^{(\tau)}} [l_{m,h,g^{(\tau)}}(s,a,s',g)] - \langle B_h(g)v_{m,h}(g) - B_h(\widetilde{g})v_{m,h}(\widetilde{g}), X_{m,h}(g^{(\tau)}) \rangle \right)^2 \right\}.$$

Notice that $\widetilde{g}^{(g)}$ depends on $g$ and in the rest of this paper we use $\widetilde{g}$ for simplicity of notation. With features $(v_{1:m,h}^{(g)}, B_h^{(g)}, X_{1:m,h}^{(g)})$, the confidence set $\mathcal{G}^{(t)}$ is then given as the collection of all possible hypothesis $g$ such that the sum of squares of the bilinear forms across the available batch data is not

greater than a pre-defined parameter $R^2$:

$$\mathcal{G}^{(t)} = \left\{ g \in \mathcal{G} : \sum_{\tau=1}^{t-1} \sum_{m=1}^{M} \left( \langle B_h^{(g)}(g) v_{m,h}^{(g)}(g) - B_h^{(g)}(\widetilde{g}) v_{m,h}^{(g)}(\widetilde{g}), X_{m,h}^{(g)}(g^{(\tau)}) \rangle \right)^2 \leq 4R^2 \right\}. \quad (5)$$

Due to the definition of $(v_{1:m,h}^{(g)}, B_h^{(g)}, X_{1:m,h}^{(g)}, \widetilde{g})$, the bilinear form

$$\langle B_h^{(g)}(g) v_{m,h}^{(g)}(g) - B_h^{(g)}(\widetilde{g}) v_{m,h}^{(g)}(\widetilde{g}), X_{m,h}^{(g)}(g^{(\tau)}) \rangle$$

approximates $\langle W_{m,h}^*(g) - W_{m,h}^*(g^*), X_{m,h}^*(g^{(\tau)}) \rangle$ for all $g \in \mathcal{G}$. Therefore the confidence set $\mathcal{G}^{(t)}$ contains all the hypothesis in which the sum of $\left( \langle W_{m,h}^*(g) - W_{m,h}^*(g^*), X_{m,h}^*(g^{(\tau)}) \rangle \right)^2$ over all history data and tasks is upper bounded by $R^2$. We will show that this can be used to quantify the Bellman error of the candidate action-value functions under the corresponding greedy policy, which then can be related to the sub-optimality of the corresponding greedy policy in the true environment. The width parameter $R^2$ can be enlarged to handle the cases where realizability assumption is not strictly satisfied. Since ERM is robust to misspecification, optimism is still guaranteed and the analysis follows similarly. The resulting algorithm would thus have sub-optimality depending additively on the misspecification error.

This confidence set captures the shared information $B_h(\cdot)$ across tasks and contains all hypothesis with low Bellman errors within a smaller $R^2$. Algorithm 1 then samples trajectories according to the greedy policies of the chosen hypothesis $g^{(t)}$ and the estimation policy $\pi_{\text{est}}$ and augments the available batch data with $\mathcal{D}_{m,h}^{(t)}, h \in [H], m \in [M]$.

We note that the main computation workload of Algorithm 1 is the ERM step in Line 3. This means that Algorithm 1 is oracle-efficient with access to an ERM oracle. In general, Line 3 pertains to a difficult optimization problem over the set of candidate function approximators $\mathcal{G}^{(t)}$. Although no efficient algorithms are currently known to solve this problem, we note that for certain instances of the bilinear class (e.g., Yang et al. (2021)) where $\mathcal{G}$ is parameterized by variables in a Euclidean space, computationally efficient gradient-based algorithms may exist to solve Line 3.

## 5 MAIN THEORY

This section presents the theoretical analysis of Algorithm 1. Without loss of generality, we assume that the feature classes are all bounded.

**Definition 5.1.** Assume $\|v_{m,h}(g)\|_2, \|B_h(g)\|_F \leq C_W$ and $\|X_{m,h}(g)\|_2 \leq C_X$ hold for all $v_{m,h} \in \mathcal{V}_h, B_h \in \mathcal{B}_h, X_{m,h} \in \mathcal{X}_h$ and $g \in \mathcal{G}, h \in [H], m \in [M]$. Here $C_W, C_X \in \mathbb{R}$.

Next, it is important to consider the expressiveness of function classes $\mathcal{V}, \mathcal{B}, \mathcal{X}$. The following assumption is common in the theory of reinforcement learning Jin et al. (2020b); Wang et al. (2020b); Jin et al. (2021b); Du et al. (2021).

**Assumption 5.2** (Realizability). Assume $g^* \in \mathcal{G}$ and $v_{1:M,1:H}^* \in \mathcal{V}, B_{1:H}^* \in \mathcal{B}, X_{1:M,1:H}^* \in \mathcal{X}$.

Now we present the theory for Algorithm 1.

**Theorem 5.3.** Set $T = 8HMd \log(1 + \frac{MC_X^2}{\lambda})$ and

$$R^2 = 8H^3 \left( Mk \log(Mn_0 C_W C_X) + \log\left( \frac{HT|\mathcal{G}||\mathcal{V}||\mathcal{B}||\mathcal{X}|}{\delta} \right) \right) / n_0.$$

With probability $1 - \delta$ the algorithm outputs a set of policies $\pi^{g^{(t_0)}}$ such that

$$\sum_{m=1}^{M} V_{m,1}^{\pi_m^*}(s_1) - \sum_{m=1}^{M} V_{m,1}^{\pi_{m,g^{(t_0)}}}(s_1) \leq O(HR).$$

The total number of trajectories used in Algorithm 1 is upper bounded by $O(MHTn_0)$. Therefore, with probability at least $1 - \delta$, Algorithm 1 is able to use

$$O\left( \frac{H^6 M^2 d(Mk + \log(|\mathcal{X}||\mathcal{V}||\mathcal{B}||\mathcal{G}|/\delta))}{\epsilon^2} \right)$$

*trajectories to find a set of policies $\{\pi_m\}_{m \in [M]}$ such that*

$$\sum_{m=1}^{M} V_{m,1}^{\pi_m^*}(s_1) - \sum_{m=1}^{M} V_{m,1}^{\pi_m}(s_1) \leq \epsilon. \tag{6}$$

Notice that if we use Bilin-UCB to learn each single task individually, then the total number of trajectories to achieve the above guarantee is $\frac{H^6 M^3 d^2 \log(|\mathcal{G}|/\delta)}{\epsilon^2}$. Indeed, to achieve Eq. (6), the average sub-optimality of each task should be $O(\epsilon/M)$. Using Bilin-UCB, achieving $V_{m,1}^{\pi_m^*}(s_1) - V_{m,1}^{\pi_m}(s_1) \leq \epsilon/M$ costs $O(H^6 M^2 d^2 \log(|\mathcal{G}|/\delta)/\epsilon^2)$ samples for each task. Thus the total number of trajectories is $\frac{H^6 M^3 d^2 \log(|\mathcal{G}|/\delta)}{\epsilon^2}$ via learning each task individually.

Therefore, Theorem 5.3 improves the sample complexity of learning Low-rank Multi-task Bilinear class given small sizes of expressive feature classes $\mathcal{V}, \mathcal{B}, \mathcal{X}$, for example, $\log(|\mathcal{X}||\mathcal{V}||\mathcal{B}|) \leq Md$.

In general, the dependence on $d$ can not be reduced since we estimate a d-by-k matrix for shared representation. Furthermore, we believe that without further assumption, a polynomial improvement w.r.t $M$ is impossible because the learner has to learn the task-specific features $v_{m,h}^* \in R^k$ for all $m \in [M]$.

For an important special case known as Linear Mixture Model (Definition 3.3), we have the following result via plugging $|\mathcal{V}| = |\mathcal{B}| = 1, |\mathcal{X}| = |\mathcal{G}|$ into Theorem 5.3.

**Corollary 5.4.** *Consider Low-rank Multi-task Bilinear class where each MDP $\mathcal{M}_m$ is a linear mixture model with $\mathbb{P}_{m,h}(s,a,s') = \langle \theta_{m,h}^*, \phi_m(s,a,s') \rangle, r_{m,h}(s,a) = \langle \theta_{m,h}^*, \psi_m(s,a) \rangle$ and there exist $\nu_{m,h}^* \in \mathbb{R}^k, B_h^* \in \mathbb{R}^{d \times k}$ such that $\theta_{m,h}^* = B_h^* \nu_{m,h}^*$ for all $h \in [H], m \in [M]$. Let*

$$\mathcal{X} = \mathcal{G} = \mathcal{N}(\mathbb{R}^k, \epsilon)^{\otimes MH} \otimes \mathcal{N}(\mathbb{R}^{d \times k}, \epsilon)$$

*where $\mathcal{N}(\mathbb{R}^k, \epsilon)$ denotes the $\epsilon$-covering of $\mathbb{R}^k$ and $\mathcal{N}(\mathbb{R}^{d \times k}, \epsilon)$ denotes the $\epsilon$-covering of $\mathbb{R}^{d \times k}$. Under Assumption 5.2, there exists an algorithm that with probability at least $1 - \delta$ finds a set of policies $\{\pi_m\}_{m \in [M]}$ such that*

$$\sum_{m=1}^{M} V_{m,1}^{\pi_m^*}(s_1) - \sum_{m=1}^{M} V_{m,1}^{\pi_m}(s_1) \leq \epsilon$$

*using*

$$O \left( \frac{H^6 M^2 d(Mk + kd) \log(1/\epsilon\delta))}{\epsilon^2} \right)$$

*trajectories.*

*Remark* 5.5. Using Bilin-UCB to learn each task individually, it takes $O\left(H^6 M^3 d^3 \log(1/\delta) \cdot \epsilon^{-2}\right)$ trajectories to learn a set of policies $\{\pi_m\}_{m \in [M]}$ such that

$$\sum_{m=1}^{M} V_{m,1}^{\pi_m^*}(s_1) - \sum_{m=1}^{M} V_{m,1}^{\pi_m}(s_1) \leq \epsilon.$$

Therefore, Algorithm 1 achieves sample complexity improvement comparing to single-task learning. Without further assumption, a polynomial improvement w.r.t $M$ appears impossible because the learner has to learn the task-specific features $v_{m,h} \in \mathbb{R}^k$ for all $m \in [M]$ with an $\Omega(M^3 k)$ sample complexity in total. The main benefit of multi-task learning in this case is to reduce the dependence on the ambient dimension $d$ to $k$.

## 6 TECHNIQUE OVERVIEW

This section gives an overview of the analysis and the main techniques. Owing to optimism principle and the construction of confidence set via representation learning, the proof of Theorem 5.3 will crucially depend on the following three observations:

**Risk bounds for representation learning.** Since the discrepancy measures serve as unbiased estimations of bilinear forms, i.e.

$$\mathbb{E}^m_{a_{m,1:h-1}\sim\pi_{m,f},a_{m,h}\sim\pi_{\text{est},m}(f)}[l_{m,h,f}(o_{m,h},g)] = \langle W^*_{m,h}(g) - W^*_{m,h}(g^*), X^*_{m,h}(f)\rangle,$$

the solutions $(v^{(g)}_{1:m,h}, B^{(g)}_h, X^{(g)}_{1:m,h}, \widetilde{g}^{(g)})$ of ERM will be able to concentrate to the population mean, i.e.

$$\langle B^*_h(g)v^*_{m,h}(g) - B^*_h(g^*)v^*_{m,h}(g^*), X^*_{m,h}(g^{(\tau)})\rangle \approx \langle B^{(g)}_h(g)v^{(g)}_{m,h}(g) - B^{(g)}_h(\widetilde{g})v^{(g)}_{m,h}(\widetilde{g}), X^{(g)}_{m,h}(g^{(\tau)})\rangle.$$

This means that Algorithm 1 has approximately captured the correct bilinear forms. Thus $\mathcal{G}^{(t)}$ essentially finds all hypothesis $g$ such that $\sum_\tau(\langle B^*_h(g)v^*_{m,h}(g) - B^*_h(g^*)v^*_{m,h}(g^*), X^*_{m,h}(g^{(\tau)})\rangle)^2$ is small.

**Regret decomposition associated to the bilinear forms.** One key property of Bilinear class is the upper bound of Bellman error as follow:

$$\left|\mathbb{E}^m_{a_{m,1:h}\sim\pi_{m,g,1:h}}[Q_{m,h,g}(s_{m,h},a_{m,h}) - r_{m,h}(s_{m,h},a_{m,h}) - V_{m,h,f}(s_{m,h+1})]\right|$$
$$\leq |\langle W^*_{m,h}(g) - W^*_{m,h}(g^*), X^*_{m,h}(g)\rangle|.$$

Furthermore, we know that the sub-optimality of the greedy policy of candidate hypothesis can be decomposed into the sum of Bellman errors across time steps. Therefore, we show the following upper bound

$$\sum_{m=1}^M V_{m,1,g}(s_1) - \sum_{m=1}^M V^{\pi_{m,g}}_{m,1}(s_1) \leq \sum_{m=1}^M \sum_{h=1}^H \left|\langle B^*_h(g)v^*_{m,h}(g) - B^*_h(g^*)v^*_{m,h}(g^*), X^*_{m,h}(g)\rangle\right|.$$

**Coverage condition of features $X_{m,h}(\cdot)$.** Owing to the previous two observations, the issue left is to upper bound $|\langle B^*_h(g)v^*_{m,h}(g) - B^*_h(g^*)v^*_{m,h}(g^*), X^*_{m,h}(g)\rangle|$ for some $g = g^{(t)}$ by the quantity $\sum_\tau(\langle B^*_h(g)v^*_{m,h}(g) - B^*_h(g^*)v^*_{m,h}(g^*), X^*_{m,h}(g^{(\tau)})\rangle)^2$. Using Hölder's inequality, it suffices that $\sum_\tau X^*_{m,h}(g^{(\tau)})X^*_{m,h}(g^{(\tau)})^\top$ possess sufficient coverage condition of $X^*_{m,h}(g)$ for all $m \in [M]$. Setting $T = \Omega(Md)$, we will confirm this via elliptical potential lemma.

The proof of Theorem 5.3 is then established based on the above three observations. Since the second and third steps are natural extensions of the analysis of Bilin-UCB, the technical bulk is then to build risk bounds for representation learning. Specifically, we define the following failure event:

**Definition 6.1.** Define $\mathcal{E}$ as the event that there exist $t \in [T]$ and $h \in [H]$ such that

$$\sum_{\tau=1}^{t-1}\sum_{m=1}^M \left(\langle B^*_h(g)v^*_{m,h}(g) - B^*_h(g^*)v^*_{m,h}(g^*), X^*_{m,h}(g^{(\tau)})\rangle\right.$$
$$\left. - \langle B^{(g)}_h(g)v^{(g)}_{m,h}(g) - B^{(g)}_h(\widetilde{g})v^{(g)}_{m,h}(\widetilde{g}), X^{(g)}_{m,h}(g^{(\tau)})\rangle\right)^2 \geq R^2.$$

We will show that $\mathbb{P}[\mathcal{E}] \leq 1 - \delta$ with the choice of $R^2$ in Algorithm 1. The key step is embedding the estimation noise into low dimensional space $\mathbb{R}^k$ and achieve improved concentration.

## 7 CONCLUSION

This paper considers learning multiple RL problems jointly with representation learning. A structural framework is proposed that permits generalization across tasks via general function approximation. A sample efficient algorithm is designed based on representation with ERM and optimistic principle where the confidence sets are constructed based on learned features. Theoretical analysis is displayed that the algorithm finds nearly-optimal policies within limited samples. Several examples are discussed and sample complexity improvements are illustrated.

Given the success of representation learning in multi-task RL, it is an interesting future direction to study transfer learning for quickly adapting prior knowledge to a new, unseen task with limited data and computational power.

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
