# OpenReview forum: "Provably efficient multi-task Reinforcement Learning in large state spaces"
_ICLR.cc/2023/Conference — Submitted to ICLR 2023_

### Official Review · Reviewer_7kqp · 2022-10-24

**Confidence:** 3
**Correctness:** 4
**Technical Novelty And Significance:** 2
**Empirical Novelty And Significance:** Not applicable
**Recommendation:** 5

**Clarity, Quality, Novelty And Reproducibility:**

Although the notation is a little overwhelming in the formulation of the ERM problem in particular, the work is reasonably straightforward and so it is reasonably clear.

As mentioned above, the novelty seems to lie primarily in being the first to combine the two notions of bilinear classes of RL environments with feature-sharing multi-task learning. This is OK, but a bit limited.

As far as quality goes, the work seems sound but (as mentioned above) obtains weaker results than I'd expect. Proofs are included and so reproducibility is not at issue.

**Strength And Weaknesses:**

The main strength is that the work extends the current state of the art in the theory of RL in large feature spaces by allowing multiple tasks to share features in a natural way. The work seems sound, and fairly general.

There are three main weaknesses: first, given that one has set out to consider multi-task + RL, the work has some flavor of "turning the crank," combining known approaches to feature sharing in multi-task learning with the techniques for analyzing RL from recent works.

Second, from this perspective, the bounds obtained are surprisingly weak, in that they retain a high-order dependence on Md, the number of tasks times the ambient feature dimension. In this setting, I would expect a dependence on Mk with a lower-order dependence on d (that is independent of the number of tasks). No justification is provided for why such a bound may not be possible, leaving the impression that this bound is a "first stab."

Third, the approach relies on an ERM oracle. While this limitation is shared with many works in RL, some recent works on linear classes (e.g., Jin et al., Yang & Wang) do not require this. The work hints that an efficient algorithm might exist for these classes, but stops short of providing one. The work would have been much more compelling if such an algorithm had been described.

**Summary Of The Paper:**

This paper provides bounds on the multi-task sample complexity of reinforcement learning in bilinear classes (recently introduced by Du et al., generalizing several existing notions) when the Bellman errors of the tasks can be parameterized in terms of a small set of common features.

**Summary Of The Review:**

On the one hand, I believe that the bounds claimed have not appeared previously in the literature. On the other hand, there does not seem to be much technical novelty involved in obtaining these bounds. The bounds seem likely to be suboptimal and the method is not computationally efficient.

---

> ### Author Response · Authors · 2022-11-18
> **Response**
>
> Thank you very much for your valuable feedback!
>
> >given that one has set out to consider multi-task + RL, the work has some flavor of "turning the crank," combining known approaches to feature sharing in multi-task learning with the techniques for analyzing RL from recent works.
>
>
>
> - We would like to comment that existing approaches in (multi-task) RL are not sufficient to address the general structure in the setting of multi-task bilinear class we discussed. Indeed, the main technical improvements are learning common representations via ERM and transferring to the in-distribution Bellman residue of target policies for multiple tasks. Applying known approaches to construct the confidence region would result in a scaling of $Md$ in the statistical error and thus fail to exploit the low-dimensional structures. This is intrinsic to the structure of the bilinear class and is also the reason why the original work Du et al. (2021) did not achieve optimal dependence in the effective dimension. These challenges are addressed by Lemma B.1 and Lemma B.4 as different from Du et al. (2021).
>
>
>
>
> >the bounds obtained are surprisingly weak, in that they retain a high-order dependence on Md
>
>
>
> -   In general, the dependence on $d$ can not be reduced since the shared representation is a $d$-by-$k$ matrix. Without representation learning, the sample complexity would scale quadratic in $d$ (cf. Jin et al. (2018) and Du et al. (2021)). Our algorithm is able to reduce it to depend linearly on d.
>
> -   Furthermore, we believe that without further assumption, a polynomial improvement w.r.t $M$ is impossible because the learner has to learn the task-specific features $v_{m,h} \in R^k$ for all $m \in [M]$. This would induce an $\Omega(M^3k)$ sample complexity in total.
>
> -   The main benefit of multi-task learning is to reduce the dependence of the statistical error from the ambient dimension $d$ to the actual dimension $k$ of task-specific representations. Our algorithm enjoys a factor of sample complexity reduction of roughly $1 − (Mk+kd)/Md$ (e.g. in linear mixture model) compared with learning each task individually, which is non-trivial when M ≫ k and becomes more significant as the number of tasks gets larger.
>
>
>
>
>
>
> >The work hints that an efficient algorithm might exist for these classes, but stops short of providing one. The work would have been much more compelling if such an algorithm had been described.
>
>
>
> - Thank you for your valuable suggestions. It is indeed a very interesting future direction to design computation efficient and sample efficient algorithms for multi-task RL with general function approximation. We would like to clarify that we intend to illustrate the existence of certain cases where computationally efficient algorithms may exist instead of asserting the existence of such an algorithm for general classes. For example, in the linear bandit setting where the actions are from $d$-dimensional Euclidean space, the solution to the ERM and the confidence region are analytically accessible, and thus our algorithm can be implemented efficiently. We have modified the related discussions of computational efficiency.

---

### Official Review · Reviewer_C4vX · 2022-10-24

**Confidence:** 2
**Correctness:** 3
**Technical Novelty And Significance:** 3
**Empirical Novelty And Significance:** 2
**Recommendation:** 5

**Clarity, Quality, Novelty And Reproducibility:**

Clarity:
The paper is well-written and the analysis of the related work is appropriate.

Novelty:
I do not feel confident to evaluate its novelty.

Reproducibility:
No validation of the proposed algorithm in any simulated or real problem, therefore, it is not easy to evaluate the reproducibility of the algorithm.

**Strength And Weaknesses:**

Strengths:
The proposed representation learning in low-rank multi-task bilinear task seems to be a very interesting idea. However, I do not feel confident to evaluate its novelty.

Weaknesses:
The proposed algorithms are not validated on any RL problem, simulated or real. For me this is mandatory. I suggest to include an experimental validation, even in a toy problem.

**Summary Of The Paper:**

This paper studies the multi-task learning problem with representation learning. Specifically it proposes  online representation learning algorithms to capture the shared features in the different task-specific bilinear forms. Unfortunately, the proposed algorithms are not validated on any RL problem, simulated or real.

**Summary Of The Review:**

The proposed representation learning in low-rank multi-task bilinear task seems to be a very interesting idea. However, the proposed algorithm is not validated in any RL problem. My opinion is that such a validation is mandatory.

---

> ### Author Response · Authors · 2022-11-18
> **Response**
>
> Thank you very much for your valuable feedback!
>
> > I suggest to include an experimental validation, even in a toy problem.
>
>
>
> -   Thank you for your valuable suggestions. Our work mainly makes a theoretical contribution that is the first to achieve sample complexity improvement in more general settings of multi-task RL by a novel construction of the confidence region via ERM. We plan to add empirical results, especially experiments on multi-task chain MDP in later updates.
>
> -   We don't want to argue the practicality of our algorithm. Instead, our algorithm is in some sense information theoretical -- our work identifies a statistically tractable subclass of MDP models and the algorithm serves as a certificate for the regret upper bound. Identifying tractable MDP classes is pivotal as the challenge of exploration is different from standard supervised learning. As a result, the classical complexity notions such as VC dimension cannot capture the hardness of exploration.

---

### Official Review · Reviewer_k93P · 2022-10-25

**Confidence:** 2
**Correctness:** 4
**Technical Novelty And Significance:** 3
**Empirical Novelty And Significance:** 3
**Recommendation:** 8

**Clarity, Quality, Novelty And Reproducibility:**

The paper was overall well written however I found the notation used in the paper a bit hard to follow due to the numerous subscripts.

**Strength And Weaknesses:**

Strengths:

_ this paper tackles an important challenge multi task learning in reinforcement. As far as I know this paper present the first result for provably efficient multi task reinforcement learning with function approximation
_ the PAC bound for theorem 5.2 shows the benefit of multi task learning is that it is more efficient than learning the different tasks seperately

Weaknesses:

_ in the definition of low rank multi-task bilinear class the paper the features W still depend on m, in practice we often try to use a single representation to solve multiple MDPs / tasks. I understand that to provide a learning bound it is necessary to have more flexibility in the representation so each tasks can be solved using the shared representation. Nevertheless could you expand a bit on the relationship between you work and using a single representation for all tasks?
_ it would be useful to provide simple examples of the algorithm on a few simple tasks to assess its empirical performance. Moreover these experiments might highlight issues of the learning dynamics of multi-task reinforcement learning such as ray interference [1]
_ the realizability asumption (assumption 5.2) seems reasonable to provide a theoretical result but it may not be verified in practice? Do you have an intuition on the performance of the proposed algorithm when this assumption is not verified?
_ the benefits of the proposed algorithm is only achieved when k << d. It would be useful to provide a sense of how often this condition is verified.

[1] Ray Interference: a Source of Plateaus in Deep Reinforcement Learning, Schaul et al., 2019

**Summary Of The Paper:**

This paper tackles the challenge of learning an efficient representation for multi task reinforcement learning. The authors present a learning algorithm for multi-task RL with function approximation, low-rank multi task bilinear class. This algorithm incentivizes the the RL algorithm to improve sample efficiency by leveraging information from multiple tasks at the same time.
The authors provides a PAC bound showing that their algorithm is more efficient than learning each task separately.

**Summary Of The Review:**

This paper provides the first provably efficient multi task reinforcement learning algorithm. As such I think it improves our understanding of representation learning and multi task learning, for that reason I recommend accepting the paper.
Yet I recommend that the authors add a few sections with empirical results and discuss more the validity of the assumptions used in their proof.

---

> ### Author Response · Authors · 2022-11-18
> **Response**
>
> Thank you very much for your valuable feedback!
>
>
>
> >Could you expand a bit on the relationship between your work and using a single representation for all tasks?
>
>
>
> -   Indeed, a great number of multi-task/meta RL algorithms use a single representation to solve multiple tasks due to specific structures in the MDPs that allows for efficient approximation of certain task-invariant features. However, in many interesting instances of multi-task bilinear classes (cf Appendix A), the features W’s still contain task-specific components. For example, the settings studied in Yang et al. (2021) and Hu et al. (2021) can be seen as special cases of multi-task bilinear class where the features W’s depend on particular tasks. As a result, using a single representation for the feature W is not sufficient to achieve PAC learning in the general setting that we discuss.
>
> -   That being said, it is important to note that our algorithm reduces to using a single representation for the feature W in many special cases: for example, when the feature $v^*_{m,h}$ is trivial (cf. the settings in recent works Cheng et al. (2022), Lu et al. (2022), and Agarwal et al. (2022)) our algorithm essentially uses the single shared feature $B^*_h$. In this sense, our algorithm can be seen as encompassing the method of using a single feature for multiple tasks.
>
> -   In conclusion, we believe that it is essentially the structural assumption of the MDPs that decides whether a single representation should be used for all tasks. We choose to let the function approximators W’s to depend on specific tasks because we would like to cover as more general multi-task RL problems as possible.
>
>
>
>
> >Do you have an intuition on the performance of the proposed algorithm when this assumption is not verified?
>
>
>
> - The width parameter $R^2$ can be enlarged to handle the cases where realizability assumption is not strictly satisfied. Since ERM is robust to misspecification, optimism is still guaranteed and the analysis follows similarly. The resulting algorithm would thus have sub-optimality depending additively on the misspecification error.
>
>
>
> >the benefits of the proposed algorithm is only achieved when k << d. It would be useful to provide a sense of how often this condition is verified.
>
>
>
> -   The case when k << d is natural because it is the same case studied in previous multi-task bandit papers. Furthermore, it is general and covers many existing models such as Hu et al. (2021) and Yang et al. (2021). This is also the meaningful setting where multi-task RL algorithms may benefit from learning the shared representation.
>
> -   This condition is often satisfied when the transition dynamics of the MDPs possess shared structures. For example, in latent variable models where the transition from the current state and deployed action to the latent state is shared across the MDPs, the condition k << d is usually satisfied since the dimension of the latent states is much smaller than the dimension of the ambient state space.
>
>
>
>
> >Yet I recommend that the authors add a few sections with empirical results and discuss more the validity of the assumptions used in their proof.
>
>
>
> - Thank you for your valuable suggestions. Realizability assumption is a common modeling assumption in RL since approximation error is not something one can prove is small and thus we only compare against the best-in-class. It is also necessary in the sense that PAC learning is not possible if the ground truth is not approximated by the function class. The multi-task bilinear setting is by far one of the most general frameworks to address multi-task RL problems and encompasses many interesting special cases. Therefore we believe that the validity of the assumptions in the proof is usually guaranteed.
> - We don't want to argue the practicality of our algorithm. Instead, our algorithm is in some sense information theoretical -- our work identifies a statistically tractable subclass of MDP models and the algorithm serves as a certificate for the regret upper bound. Identifying tractable MDP classes is pivotal as the challenge of exploration is different from standard supervised learning. As a result, the classical complexity notions such as VC dimension cannot capture the hardness of exploration. We plan to add empirical results, especially experiments on multi-task chain MDP in later updates.

---

### Decision · Program_Chairs · 2023-01-20

**Decision:**

Reject

**Justification For Why Not Higher Score:**

Theoretical results of questionable significance

Authors openly acknowledge ongoing empirical work suggesting work in progress not yet ready for publication

**Justification For Why Not Lower Score:**

N/A

**Metareview: Summary, Strengths And Weaknesses:**

This paper presents progress towards a sample efficient, multi-task Reinforcement Learning (RL) algorithm. Some theoretical results are proven, but the bounds proven are weak with little novelty beyond the application to multi-task RL. The significance of this contribution could be improved by tighter bounds or empirical evaluation of a concrete algorithm. The authors acknowledge they are planning to add empirical results in future updates suggesting the work is still in progress. Overall, this appears to be an intermediate result in an important research direction. Hopefully the feedback from all reviewers will help to improve the authors' future work on this topic.